# Monitoring SARS-CoV-2 Populations in Wastewater by Amplicon Sequencing and Using the Novel Program SAM Refiner

**DOI:** 10.3390/v13081647

**Published:** 2021-08-19

**Authors:** Devon A. Gregory, Chris G. Wieberg, Jeff Wenzel, Chung-Ho Lin, Marc C. Johnson

**Affiliations:** 1Department of Molecular Microbiology and Immunology, University of Missouri, Columbia, MO 65201, USA; gregoryde@missouri.edu; 2Water Protection Program, Missouri Department of Natural Resources, Jefferson City, MO 65102, USA; chris.wieberg@dnr.mo.gov; 3Bureau of Environmental Epidemiology, Missouri Department of Health and Senior Services, Division of Community and Public Health, Jefferson City, MO 65102, USA; Jeff.Wenzel@health.mo.gov; 4Center for Agroforestry, School of Natural Resources, University of Missouri, Columbia, MO 65201, USA; LinChu@missouri.edu

**Keywords:** coronavirus, wastewater, metagenomics, molecular epidemiology

## Abstract

Sequencing Severe Acute Respiratory Syndrome Coronavirus 2 (SARS-CoV-2) from wastewater has become a useful tool in monitoring the spread of viral variants. Approaches to this task have been varied, relying on differing sequencing methods and computational analyses. We used a novel computation workflow based on amplicon sequencing of SARS-CoV-2 spike domains in order to track viral populations in wastewater. As part of this workflow, we developed a program, SAM Refiner, that has a variety of outputs, including novel variant reporting as well as functions designed to remove polymerase chain reaction (PCR) generated chimeric sequences. With these methods, we were able to track viral population dynamics over time. We report here on the emergence of two variants of concern, B.1.1.7 (Alpha) and P.1 (Gamma), and their displacement of the D614G B.1 variant in a Missouri sewershed.

## 1. Introduction

Severe Acute Respiratory Syndrome Coronavirus 2 (SARS-CoV-2) caused a pandemic and a worldwide health crisis starting in 2020 [1]. Full genome sequences of SARS-CoV-2 were rapidly made available within the first months of spread [2,3]. Partial- and whole-genome sequencing of SARS-CoV-2 have been important tools in monitoring transmission paths and the emergence of variant lineages. Sequencing of SARS-CoV-2 has mostly been performed using clinical samples. However, early in the SARS-CoV-2 pandemic, wastewater was used to track community levels and spread of SARS-CoV-2 by Reverse Transcription-Quantitative Polymerase Chain Reaction (RT-qPCR) methods [4,5]. Investigators have also used high-throughput sequencing on wastewater samples to obtain full or partial SARS-CoV-2 genomic sequences which have been used for metagenomic and epidemiologic analysis [6,7,8,9,10,11,12,13]. Sequences identified in wastewater samples may reflect known lineages as well as lineages not reported from clinical samples. Combinations of mutations not observed in clinical samples may represent new infections not yet picked up by clinical sampling or lineages that are under-represented in clinical samples. Approaches using wastewater are particularly relevant with the emergence of variant lineages that may vary from previous isolates in their fitness and/or pathogenesis.

The state of Missouri has been monitoring wastewater to track the prevalence and spread of SARS-CoV-2 using RT-qPCR (https://storymaps.arcgis.com/stories/f7f5492486114da6b5d6fdc07f81aacf accessed on 23 June 2021). We sought to begin using the same samples for high-throughput sequencing to track the presence and spread of known and previously unreported variant lineages. We were specifically interested in the spike gene, so we used primers to target 3 regions for amplification, the N-terminal domain (NTD), the receptor binding domain (RBD) and the region of the S1 and S2 subunit split (S1S2). We chose these regions due to the numerous variations matching evolving lineages found in them and their significance in potential immune evasion [14]. While there are a number of high-throughput sequencing technologies and methods, the sequencing output is relatively standard, whereas the processing and analysis of that sequence data are not. There are numerous programs and pipelines that can be used to obtain information from sequences and remove errors generated from PCR, such as single-nucleotide (nt) polymorphisms (SNPs) and chimeric sequences. While many of these are quality approaches, we were unable to find a simple program or workflow with existing programs for high-throughput sequencing data that produced a condensed report of known and unknown co-variants found in the data. We wanted the variant report to include SNPs, multiple nucleotide polymorphisms (MNPs), insertion and deletion events (indels), and downstream amino acid changes, and also wanted PCR-generated chimeric sequences removed. While some programs or pipelines partially fulfilled these criteria, none we found did so fully.

Here, we detail the workflow we used to analyze high-throughput sequencing data and the program we developed to provide a human-readable, information-dense output for viewing lineages that meet the criteria described above. Using this workflow and our program, we were able to monitor SARS-CoV-2 population changes in a Missouri sewershed.

## 2. Materials and Methods

### 2.1. Wastewater Collection

Twenty-four-hour composite samples were collected at wastewater treatment facilities (WWTF) and maintained at 4 °C until they were delivered to the analysis lab, generally within 24 h of collection. Samples reported in this study were collected at the NPSD Interim Saline Creek Regional WWTF in Fenton, MO, USA.

### 2.2. RNA Extraction

Wastewater samples were centrifuged at 3000× *g* for 10 min and then filtered through a 0.22 µM polyethersolfone membrane (Millipore, Burlington, MA, USA). Approximately 37.5 mL of wastewater was mixed with 12.5 mL solution containing 50% (*w*/*vol*) polyethylene glycol 8000 and 1.2 M NaCl, mixed, and incubated at 4 °C for at least 1 h. Samples were then centrifuged at 12,000× *g* for 2 h at 4 °C. Supernatant was decanted and RNA was extracted from the remaining pellet (usually not visible) with the QIAamp Viral RNA Mini Kit (Qiagen, Germantown, MD, USA) using the manufacturer’s instructions. RNA was extracted in a final volume of 60 µL.

### 2.3. Sequencing

The primary RT-PCR (25 µL) was performed with 5 µL of RNA extracted from wastewater samples with loci-specific primers (0.5 µM each) (Table 1) using the Superscript IV One-Step RT-PCR System (Thermo Fisher, Waltham, MA, USA). Primary RT-PCR amplification was performed as follows: 25 °C(2:00) + 50 °C(20:00) + 95 °C(2:00) + [95 °C(0:15) + 55 °C(0:30) + 72 °C(1:00)] × 25 cycles. Secondary PCR (25 µL) was performed using 5 uL of the primary PCR as template with gene-specific primers containing 5′ adapter sequences (0.5 µM each), dNTPs (100 µM each) and Q5 DNA polymerase (NEB, Ipswich, MA, USA). Secondary PCR amplification was performed as follows: 95 °C(2:00) + [95 °C(0:15) + 55 °C(0:30) + 72 °C(1:00)] × 20 cycles. A tertiary PCR (50 µL) was performed to add adapter sequences required for Illumina cluster generation with forward and reverse primers (0.2 µM each), dNTPs (200 µM each), and Phusion High-Fidelity DNA Polymerase (1U). PCR amplification was performed as follows: 98 °C(3:00) + [98 °C(0:15) + 50 °C(0:30) + 72 °C(0:30)] × 7 cycles + 72 °C(7:00). The amplified product (10 µL) from each PCR reaction is combined and thoroughly mixed to make a single pool. Pooled amplicons were purified by addition of Axygen AxyPrep MagPCR Clean-up beads in a 1.0 ratio to purify final amplicons. The final amplicon library pool was evaluated using the Agilent Fragment Analyzer automated electrophoresis system, quantified using the Qubit HS dsDNA assay (Invitrogen, Waltham, MA, USA), and diluted according to Illumina’s standard protocol. An Illumina MiSeq instrument was used to generate paired-end 300 base pair length reads. Adapter sequences were trimmed from output sequences using cutadapt [15]. The raw and trimmed reads for the samples used in this report are available at https://github.com/degregory/SR_manuscript/tree/master/Fenton_Data accessed on 23 June 2021. Raw reads for all of Missouri wastewater monitoring will be available under BioProject PRJNA748354.

## 3. Results

### 3.1. Computational Pre-Processing

Figure 1 illustrates the steps of our workflow. The two steps following read trimming used the VSEARCH tool [16]. First, the trimmed paired reads were merged using vsearch ‘--fastq_merge’ with default parameters. Merged reads were then dereplicated using vsearch ‘--derep_fulllength’ with the arguments ‘--minsize 100’ and ‘--sizeout’. These arguments limit the output to unique sequences that occur at least 100 times and appends the sequence IDs with ‘size=#’, where # is the number of times that particular sequence occurred in the reads. The cutoff of 100 counts removes late-stage PCR errors, leaving only sequences representing the original templates or errors that occurred in early cycles of the PCR. This removal makes further analysis simpler and faster. However, very low frequency original template sequences will also be removed by this cutoff, so this step could be skipped to preserve such rare sequences. The resulting unique sequences were mapped to the sequence of SARS-CoV-2 (NCBI Reference Sequence: NC_045512.2, https://www.ncbi.nlm.nih.gov/nuccore/NC_045512, accessed on 7 February 2021) spike ORF using Bowtie2 [17] with default parameters to generate standard SAM formatted files. Having SAM formatted files allows the use of the program we developed for amplicon sequencing results. All files associated with these steps for our analysis of the Fenton, MO sewershed in this manuscript can be accessed at https://github.com/degregory/SR_manuscript/tree/master/Fenton_Data accessed on 23 June 2021.

### 3.2. SAM Refiner: SAM Processing

Our program, SAM Refiner, is currently a command line-based python script and is available at https://github.com/degregory/SAM_Refiner accessed on 23 June 2021 along with updated documentation. In order to run SAM Refiner, a python compiler or interpreter is needed (https://docs.python.org/3/tutorial/interpreter.html accessed on 23 June 2021). Though only tested in a Linux environment, it should function with other common operating systems. Figure 2 shows the command line usage for SAM Refiner. Standard SAM formatted files are the starting point for our program. These files are generated by many mapping programs, including Bowtie2 [17] and BWA [18]. The default functions of SAM Refiner follow. Files with the extension .sam (case insensitive) in the working directory will be identified and processed. To process SAM files, SAM Refiner must be provided a FASTA formatted file for a reference sequence using the command line argument ‘–r reference.fasta’, where the FASTA file contains the same sequence ID and sequence used to map the sequencing reads in the SAM formatted file. If the IDs of the given reference and the reference of mapped sequences in the SAM file do not match, those sequences will be ignored. If the SAM formatted files were generated from dereplicated or collapsed sequences that contain the unique read count in sequence ids where the count is at the end of the id and denoted with a ‘=’ or ‘−’, SAM Refiner will recognize the counts, i.e., ‘Seq1:1;counts = 20′ will be recognized as a sequence with 20 occurrences.

For each SAM file, SAM Refiner initially outputs 4 tab separated value (TSV) files that can be read by any standard spreadsheet software. For a SAM file with the name Sample.sam, the outputs are named Sample_unique_seqs.tsv, Sample_nt_calls.tsv, Sample_indels.tsv and Sample_covars.tsv. Example outputs of each are provided in Appendix A, respectively (https://github.com/degregory/SR_manuscript/tree/master/Supplementals accessed on 23 June 2021). All reports are based on the FASTA reference relative to the SAM formatted file, so any errors made by the mapping or incongruence between the FASTA reference and the mapping reference will result in propagated errors. The reports also include the coded amino acids and their position in the coded peptide as if the reference is an in-frame coding sequence. If multiple nucleotides in a single codon differ from the reference, they will be reported together as a MNP with the associated amino acid change. Within the files, all of the sample-specific outputs start with the name of the sample taken from the SAM file name followed in parenthesis by the count of reads mapped.

The Sample_unique_seqs.tsv file (Appendix A) lists the unique sequence reads mapped in the SAM file using a variant notation to list the variations from the reference along with occurrence count and abundance. For example, using the previously mentioned SARS-CoV-2 spike ORF as the reference sequence, a sequence read that matches the reference except for having a T at position 1501 instead of the reference A would be reported simply as ‘1501A(N501Y)’. The abundance reported uses decimal notation, so 0.2 represents 20% abundance. Unique sequences that have an abundance below 0.001 are not reported.

The Sample_nt_calls.tsv file (Appendix A) has a line for each nt position covered in at least 0.1% of the reads. Based on the reference sequence, each line first reports the nt position, the reference nt, the encoded amino acid position, and the amino acid residue encoded by the reference sequence. The line then reports the number of calls for each base and for deletions at that position, followed by the most abundant (primary) call and its counts and abundance. If the primary nt is different from the reference sequence, the amino acids encoded by the primary nt sequence and by the reference sequence with only that nt changed are reported. Further, if the second (secondary) and third (tertiary) most abundant nts are above 0.1% of the total read counts, those nts, their counts, abundances, and associated amino acid changes are also reported.

The Sample_indels.tsv (Appendix A) file lists each insertion or deletion found in the mapping along with its occurrence count and abundance. Reported insertions have the format of ‘position-insertNT(s)’, so an insertion between nt positions 54 and 55 of the sequence ‘GCA’ will be reported as ‘55-insertGCA’. Reported deletions have the format ‘start Position-end positionDel’, so a deletion of the nts at positions 61 through 64 would be reported as ‘61-64Del’. Amino acid changes are reported if the indel maintains the reading frame. If there are no indels in the reads, no indel report will be generated.

Finally, the Sample_covars.tsv (Appendix A) file lists all observed single polymorphisms and polymorphisms combinations relative to the reference sequence. The number and abundance of sequence reads containing each covariant (covar) are reported regardless of whether any of those reads have other variations or not. As an example of this processing, the sequence ‘1212G(G404G) 1501T(N501Y) 1709A(A570D)’ with 100 counts would have the covariants of ‘1212G(G404G)’, ‘1501T(N501Y)’, ‘1709A(A570D)’, ‘1212G(G404G) 1501T(N501Y)’, ‘1212G(G404G) 1709A(A570D)’, ‘1501T(N501Y) 1709A(A570D)’ and ‘1212G(G404G) 1501T(N501Y) 1709A(A570D)’, and contribute 100 counts to each. Because unique sequences that fall below the 0.1% reporting cutoff can still contribute to covariants, there may be polymorphisms in the reported covariants that are not seen in the unique sequence output. Any sequences with more than 40 polymorphisms from the reference are ignored. While all sequences with 40 or fewer polymorphisms are analyzed, only combinations of 8 or fewer polymorphisms are reported.

Once the above outputs are generated from each SAM file found, SAM Refiner will collect information from each sample and report them in a single file for the covars and unique_seqs reports (Collected_Covariances.tsv and Collected_Unique_Seqs.tsv). These collections have a threshold of 1% occurrence for reporting.

Many options are available as command line arguments that can change parameters of SAM processing of SAM Refiner (Figure 2). There are no strictly required command line arguments, though the ‘-r’ argument is required for the SAM processing. Omitting the reference sequence will cause SAM Refiner to skip SAM processing and only perform the collections and chimera removal (see below), which require pre-existing outputs. The other input option is the ‘-S’ argument, which provides SAM Refiner with SAM files to process instead of searching the working directory. The use of dereplicated/collapsed counts in the SAM files can be disabled by using ‘--use_counts 0’. There are also options available for the outputs. All outputs can be separately suppressed with the arguments ‘--seq 0’, ‘--nt_call 0’, ‘--indel 0’, ‘--covar 0’ and ‘--collect 0’. The collections file names can be prepended with a string specified by the argument ‘--colID’. To change the reporting threshold for the sample and collected outputs, arguments ‘--min_abundance1’ and ‘--min_abundance2’ are used, respectively. For ‘--min_abundance1’, despite its name, the value can be used to either set a minimal abundance threshold or a minimal count threshold. Values of 1 or greater will set a count threshold, while those less than 1 will set an abundance threshold. Only an abundance threshold is available for ‘--min_abundance2’. All amino acid information in the reports can be suppressed with the argument ‘--AAreport 0’, which is recommended if the reference does not primarily provide an in-frame coding sequence. Users can also have all nt changes processed independently, even if they are in the same codon, with ‘--AAcodonasMNP 0’. Using ‘--ntabund’ will change the required mapped coverage threshold for reporting a position in the nt_calls output. Finally, ‘--max_dist’ and ‘--max_covar’ allow changes to covar processing and reporting. Sequences with more variations than the amount specified by ‘--max_dist’ are not included in the covar analysis. The maximum number of polymorphisms reported in a combination can be set with ‘--max_covar’. As an example, if ‘--max_covar 2’ were used for Sup. 4, then ‘1216-1216Del 1501T(N501Y) 1709A(A570D)’, ‘1212G(G404G) 1501T(N501Y) 1709A(A570D)’ and ‘1217-1217Del 1501T(N501Y) 1709A(A570D)’ would not be reported.

Using the SAM files generated from the sequencing data of the Fenton sewershed, we ran SAM Refiner with the same reference as was used for Bowtie2 mapping, the SARS-CoV-2 (NCBI Reference Sequence: NC_045512.2) spike ORF sequence. The resulting outputs can be accessed at https://github.com/degregory/SR_manuscript/tree/master/Fenton_Data accessed on 23 June 2021. These outputs allow us to see the variant lineages present at different dates in this sewer shed. However, as can be seen in Appendix A, many of the sequences reported appear to be chimeric sequences arising from template jumping. While these outputs can still be used for further analysis, removing chimeric sequences makes such analysis easier, so SAM Refiner also has methods to remove such chimeric sequences.

### 3.3. SAM Refiner: Chimera Removal

PCR amplification can introduce sequence errors that obscure the original template sequences. Of most concern are the introduction of false SNPs and chimeric reads. Most PCR-introduced SNPs can be removed from analysis by the use of an abundance threshold such as is the default for SAM Refiner, or as was used in our pre-processing dereplication step. There are also numerous other programs that can be used to attempt to remove such errors. Chimeric sequences are generally more difficult to remove. Many programs exist for this task; however, we were unable to find any that provided satisfying results for our amplicon sequencing. We developed two algorithms for SAM Refiner in order to remove chimeric errors arising from PCR template jumping from the SAM processing outputs. They are redundant in their function but crosschecking between the two different methods allows for increased confidence in the results.

The algorithms to remove chimeric sequences rely on the unique sequence and covariant files generated by SAM processing. The first algorithm, chimera removed (chim rm), goes through the individual unique sequences, starting with the lowest abundance, to determine if the sequences are chimeric. Figure 3 shows a simplified example of how the determination is made on the lowest abundant sequence of an example unique sequence output (Appendix A). For this step, the sequence being considered as a potential chimera is broken up into all possible dimeric halves. Each pair is then compared to all the other sequences to detect potential parents. A sequence is flagged as a potential parent if its abundance is greater than or equal to the abundance of the potential chimera multiplied by 1.8 (foldab) and there is at least one other sequence that would be a matched parent to the complimentary dimeric half. When a pair of dimeric halves have potential parents, the abundances of parent pairs are multiplied. The products from each potential parent pairings are summed as an expected abundance value and compared to the observed abundance of the potential chimera. If the abundance of the potential chimera is less than that of the expected value multiplied by 1.2 (alpha), that sequence is flagged as a chimera and removed. The counts attributed to the flagged chimeric sequence are then redistributed to the parent sequences based on the relative expected contribution to recombination. Once this process has been performed for all the sequences, it is repeated until no more sequences are flagged as chimeric or 100 chimera removal cycles have completed. The results of this algorithm that have a recalculated abundance of 0.001 or greater are output in a new file (Appendix A Example_a1.2f1.8rd1_chim_rm.tsv). The added string represents values of the parameters used for the processing (alpha, foldab and redist; see below for more information on the parameters).

The second algorithm, covariant deconvolution (covar deconv), is a two-step process. Figure 4 shows these processes using the example outputs found in Appendix A. The first step determines if a sequence is likely to be a true or chimeric sequence by obtaining the ratio of the frequency of a given covariant sequence relative to an expected abundance of that covariant sequence assuming random recombination of its individual polymorphisms. The expected abundance is obtained by multiplying the abundances of each individual polymorphism that is present in that covariant sequence. For instance, in a sample where ‘1501T(N501Y)’ has an abundance of 0.32 and ‘1709A(A570D)’ has an abundance of 0.35, the expected abundance of the covariant ‘1501T(N501Y) 1709A(A570D)’ would be 0.112 [0.32 × 0.35]. If the ratio of the observed abundance to the expected abundance is equal to or greater than 1 (beta), that covariant passes the check and is sent to the second step. Any sequence that has an abundance of 0.3 or greater is automatically passed. If such a sequence has an observed/expected ratio less than 1, it will be assigned a ratio of 1. The second step processes the passed sequences in order of greatest observed/expected ratio to least. If multiple sequences have the same ratio, they are processed in order of greatest to least distance from the reference. Sequences that automatically pass the first step are processed after the other sequences in order of least abundant to greatest. Sequences are assigned a new occurrence count based on their constituent individual polymorphisms. For the sequence being processed, the count for the least abundant individual polymorphism is assigned to the sequence and constituent polymorphisms making up the sequence have their count reduced by the amount of the least abundant polymorphism. This reduction means the individual polymorphism that had the least counts is assigned 0 counts, so any sequence not yet processed in which that polymorphism is present is functionally removed. This process is repeated until all sequences have been reassessed or removed. The final results with an abundance of 0.001 or greater are reported in a new file (Appendix A Example_covar_deconv.tsv).

As before, the results from individual samples are collected and reported for entries above 1% occurrence. A number of command line arguments will also influence the chimera removal algorithms. Both chimera removal algorithms run by default, but either or both steps can be disabled (‘--chim_rm 0’ and ‘--covar_deconv 0’). The collections are again disabled with ‘--collect 0’. An additional output of the covariants that passed the first step of the second algorithm can be generated with ‘--pass_out 1’ (Appendix A). The outputs are constrained as before by a minimum abundance with command line arguments ‘--min_abundance1’ and ‘--min_abundance2’. Collection file names are also prepended with ‘--colID’. The only input parameter that can be changed by command line argument is the abundance of sequences or covariants that will be considered in the algorithms. By default, only entries from the inputs that have a 0.001 abundance or greater are processed. This threshold can be changed with ‘--chim_in_abund’.

Four parameters can be altered for the first algorithm. The abundance ratio that is used as a threshold for selecting potential parents of a potential chimera can be set with ‘--foldab’. Larger values will generally reduce the pool of sequences that will be considered as potential parents, thus potentially reducing the total expected abundance obtained from parent pairs and the number of sequences flagged as chimeric. In the simplest theoretical model of PCR chimera generation, two parents generate one chimera. The parents have at least twice the abundance of the chimera as they would exist and have been amplified prior to the chimera, but the reality of chimera generation can be much more complex as many sequences may generate identical chimeras multiple times. If a sample has little chimera generation, a ‘--foldab’ value close to 2, such as the default of 1.8, should be sufficient to remove chimeras without also removing non-chimeric sequences in error. However, the more chimera generation observed, the more the ‘--foldab’ value needs to be reduced to accurately remove all chimeric sequences. Though it would be rare, this value can even be set to 0 so as not to exclude any sequence from being considered a potential parent. Lower values, however, will also increase the likelihood of a sequence being flagged as a chimera in error. Users may need to empirically determine the best value for their samples.

The multiplier for the parental summed abundance for determining if a sequence is a chimera can be set with ‘--alpha’. Larger values will generally result in a greater number of sequences flagged as chimeric. As with ‘--foldab’, the optimal value for ‘--alpha’ will depend on the extent of chimera generation in the samples being processed, with a value near 1 for minimal chimera generation (such as the default 1.2) and 2 or even higher for rampant chimera generation. Once again, the later would also increase the likelihood of sequences being flagged as chimeric in error.

Redistribution of the counts from the chimera to the parent sequences can be disabled with ‘--redist 0’. Redistribution is meant to give an estimate of the counts and abundances that would have been observed without chimera generation which users may wish to forgo. The maximum number of chimera removal cycles can be changed by ‘--max_cycles’, (i.e., ‘--max_cycles 2’ will only allow two iterations of the chimera removal). Multiple removal cycles allow chimeras to be found based on new counts and abundances resulting from previous cycles, increasing the likelihood chimeras are removed from a sample.

The second algorithm has two parameters that can be changed. The ratio threshold at which a covariant will be passed to the second step can be altered with ‘--beta’. The abundance at which a covariant will automatically be passed can be changed with ‘--autopass’.

The chimera removal methods of SAM Refiner were also used on the Fenton sewershed sequencing data. Due to the relatively high amount of chimeric sequences in our samples, we used the command line arguments ‘--foldab = 0.6 –alpha = 2.2’. The outputs generated for the Fenton sewershed from 2-2-21 to 4-13-21 can be accessed at https://github.com/degregory/SR_manuscript/tree/master/Fenton_Data accessed on 23 June 2021. The two different chimera removal methods showed good concordance, validating each as being a viable detection method. Duplicate RT-PCR preparation and sequencing of the same wastewater sample also generally provided similar results, though less consistently (Figure 5. Compare A and B RBD amplicon preparations). These differences were more pronounced with covariants with relatively low abundance, such as is seen with 3-30 RBD samples, where one detects T478K and the other does not (Figure 5). These differences illustrate the stochastic nature of RT-PCR amplification.

We used the chimera removed and covariant deconvolution outputs to assign sequences to known variant lineages or the reference (Appendix A) based on polymorphisms present. Polymorphisms were considered for lineage assignment if they appeared in multiple sequencing runs or were known to be present in circulating populations reported to GSIAD (https://www.gisaid.org/, accessed on 20 February 2021). Polymorphisms that could not be validated were not taken into account for lineage assignment. Based on these assignments, we were able to observe the changes to virus populations in the sewershed over time (Figure 5). We classified the sequences found from the NTD amplicon as matching reference sequence, lineage B.1.1.7 (Alpha) with ‘203-208Del 429-431Del’ or lineage P.1 (Gamma) with ‘412T(D138Y) 570T(R190S)’ (Appendix A). Sequences from the RBD amplicon matched reference sequence, lineages B.1.1.7 with ‘1501T(N501Y) 1709A(A570D)’, P.1 with ‘1250C(K417T) 1450A(E484K) 1501T(N501Y)’, or had the single variations of T478K or L452R (Appendix A). T478K and L452R each have lineage associations. However, no other polymorphisms are associated with these in the RBD amplicons, nor were any polymorphisms present in the other amplicons that would indicate the presence of any associated lineages. While these SNPs could be the result of PCR error, it is more likely the associated lineages exist in the sewershed, but, due to stochastic effects, the other associated polymorphisms in the other amplicons were not detected. They could have also arisen in a reference background. As we cannot assign them to a known lineage with any certainty, we assigned them to their own category. Sequences from the S1S2 amplicon matched lineage B.1.1.7 with ‘1841G(D614G) 2042A(P681H) 2147T(T716I)’, lineage P.1 with ‘1841G(D614G) 1963T(H655Y) 2063T(A688V)’ or the B.1 lineage with only the now ubiquitous D614G variation (Appendix A). The 03-23 S1S2 sample had a sequence ‘1841G(D614G) 2037G(N679K) 2063T(A688V)’. While A688V is associated with P.1, it does not appear in that context here. As that is the only sample where those covariant sequences were observed and the polymorphisms are not frequently reported in GISAID (outside of P.1 for A688V), we did not feel we could validate this sequence as a novel lineage and instead tentatively assigned it to the reference category. From these results, we can conclude that the SARS-CoV-2 population of this sewershed changed in March 2021 from almost exclusively the D614G B.1 lineage to mainly the B.1.1.7 lineage, with the introduction of P.1 early in April 2021. This general method is now being used to track SARS-CoV-2 variants in many Missouri sewersheds (https://storymaps.arcgis.com/stories/f7f5492486114da6b5d6fdc07f81aacf accessed on 23 June 2021).

## 4. Discussion

### 4.1. General Discussion

Especially as new SARS-CoV-2 variants emerge that have altered viral fitness and/or pathogenesis, it is important for health professionals and policy makers to have up-to-date information on the viral populations present in communities. Surveillance of wastewater by high-throughput sequencing has proven to be a cost effective and reliable method to obtain such information [6,7,8,9,10,11,12,13]. Sequencing of wastewater for SARS-CoV-2 relies on whole-genome sequencing, targeted amplicon sequencing, or some intermediate of the two; each approach has advantages and disadvantages. Whole-genome sequencing is more likely to detect polymorphisms across the whole genome that are present in a local viral population. However, the ability to link individual polymorphisms to each other is negatively impacted by distance. The difficulty in linking polymorphisms can hinder identifying specific lineages in a population. Targeted amplicon sequencing only provides information on the targeted sequence. However, polymorphisms within the target can be easily linked and lead to easier lineage identification if the targeted sequence(s) are rich in lineage-defining polymorphisms. The spike gene, particularly the regions encoding the NTD, RBD, and S1S2 junction, is rich in such polymorphisms.

We choose these regions for targeted amplicon sequencing in order to identify lineages present in Missouri communities. This approach has proven effective in combination with our computation workflow, and we have reported here our finding for one Fenton, MO sewershed. Our results readily demonstrate the changes in this community’s viral population over time. Based on the ability to readily detect variants, our methods should also detect novel variants that have polymorphisms in these regions.

Beyond this specific application, our methods may be generalized to monitoring wastewater for variants of other viruses, virulent factors of pathogenic bacteria, human disease alleles, and many other genetic targets of interest. Aside from wastewater, our methods could also be useful in assaying other environmental samples or even clinical samples where a polymorphism rich sequence is a desirable target.

### 4.2. SAM Refiner: Limitations and Future Development

While the outputs of SAM Refiner can be very informative, the program has some limitations, some of which may be overcome in future development. Currently, the greatest limitation is the need for users to be familiar with command line usage. We hope to develop a graphical user interface version to overcome this user hurdle in the future. We also intend to develop SAM Refiner to be available from widely used functional collections such as BioConda (https://bioconda.github.io/accessed on 23 June 2021) and Galaxy (https://usegalaxy.org/accessed on 23 June 2021).

Though SAM Refiner can be used on sequencing not based on amplicons, its usefulness will be more limited in these cases as the relative abundance of sequences and covariants will be calculated based on total reads and not positional coverage. Development to include a mode for whole-genome sequencing or multiple amplicons is in process. The ability to use multiple sequences for a reference may also be added.

The accuracy of the chimera removal algorithms will vary greatly depending on the parameters used and the sample they are being run on. Due to the stochastic nature of chimera generation, and amplification during PCR, and the possible complexity of the original template sequences, samples will sometimes be refractory to chimera removal algorithms. This problem is faced by all programs designed for this purpose. The ability to modify parameters in the algorithms as well as having two algorithms with different approaches to the chimera removal may improve the accuracy the user can achieve with this software. Some samples will, however, always fail to be processed accurately by one or both methods.

## Figures and Tables

**Figure 1 viruses-13-01647-f001:**
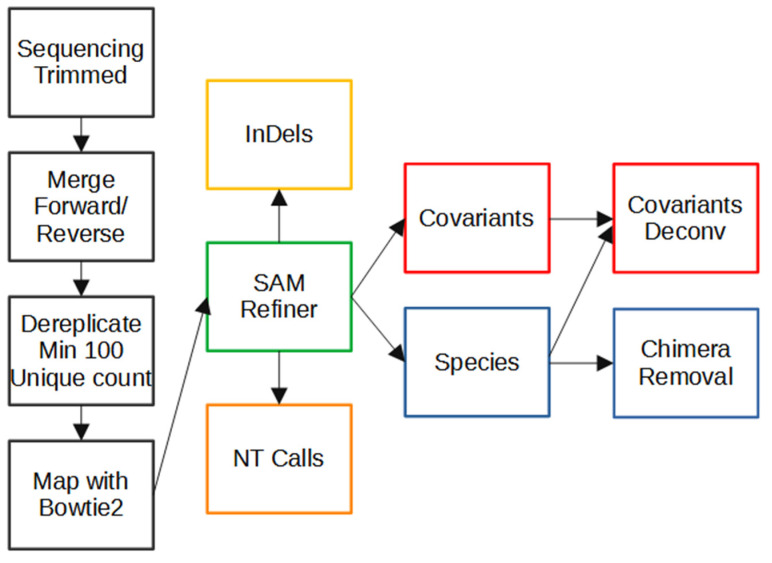
Workflow of Amplicon Sequencing Analysis. Computational processing of sequencing results prior to the use of SAM Refiner is seen in the black boxes. Paired end reads generated from an Illumina MiSeq were trimmed of low-quality calls at the end of the reads. Paired end reads were then merged into single sequence reads. Reads were then dereplicated to unique sequences with at least 100 counts while preserving the count information in the sequence IDs. Dereplicated sequences were then mapped to the sequence of the SARS-CoV-2 spike ORF using Bowtie2. SAM Refiner was then used to process the mapped reads to obtain information about the variant lineages observed, initially outputting 4 TSV files to report unique sequences, nt calls, indels and covariants. The unique sequences and covariants were further processed to remove chimeric PCR artifacts to produce covariant deconvolution and chimera removed outputs.

**Figure 2 viruses-13-01647-f002:**
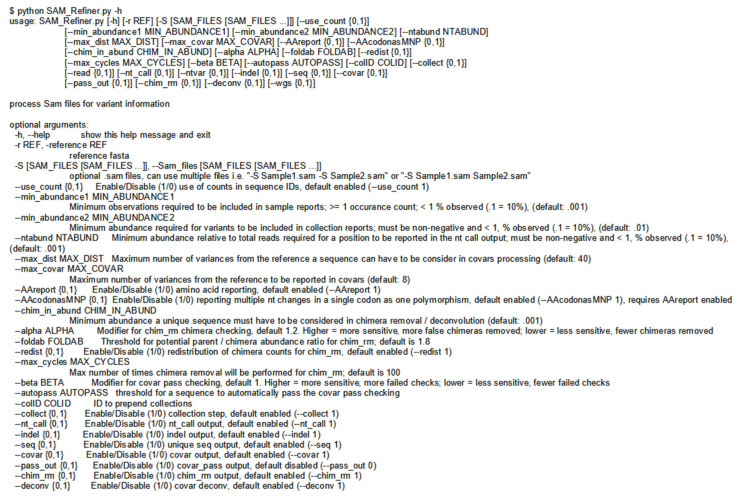
Command Line Usage of SAM Refiner. The standard help output from SAM Refiner is shown. Syntax for the command line usage is seen followed by details about potential arguments to modify program parameters.

**Figure 3 viruses-13-01647-f003:**
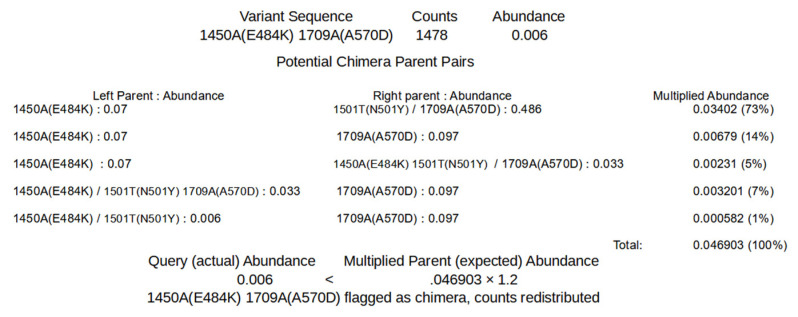
First Method of Detection and Removal of Chimeras, Chimera Removed. Using the sequences shown in Appendix A, the query of the least abundant sequence is shown. Potential parents whose recombination could result in the query sequence are found. The abundances of each potential pair are multiplied. The sum of the multiplied abundances of the pairs (expected) is then compared to the abundance of the query sequence (actual) to determine if the query sequence is a chimera. If the actual abundance is greater or equal to 1.2-fold the expected abundance, the sequence is considered non-chimeric.

**Figure 4 viruses-13-01647-f004:**
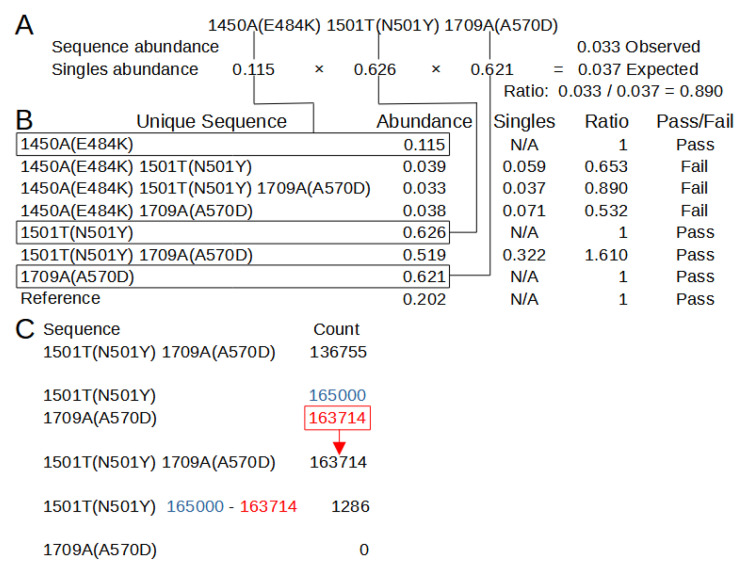
Second Chimera Removal Method in SAM Refiner, Covariant Deconvolution. (**A**) Calculations of the singles/expected abundance and abundance ratio for one of the unique sequences from Sup. 5 and the abundances from Sup. 7. Lines connect the singles and their abundance to the same in (**B**). (**B**) Calculations for determining if a unique sequence passes the initial check. Sequences pass when they have an abundance/singles ratio of 1 or greater. (**C**) Passed sequences are processed in order of greatest ratio to least. Counts of the sequence are set to the counts of the least abundant single variant, and that count is then removed from all single variants in that sequence.

**Figure 5 viruses-13-01647-f005:**
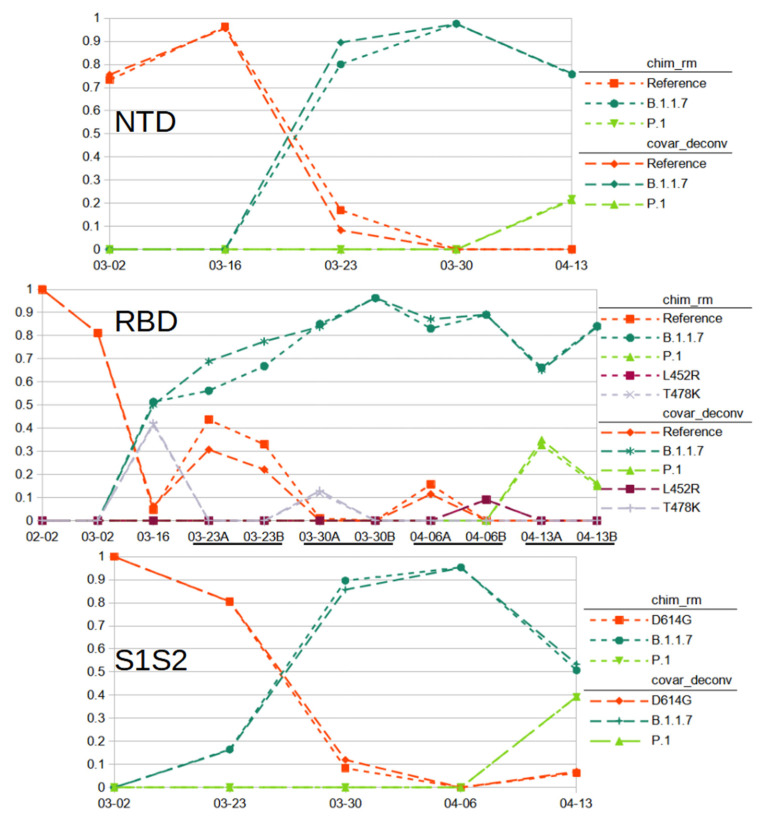
Relative Abundance of Reference and Variant SARS-CoV-2 Sequences Observed in Fenton, MO sewershed from February to March. Results from sequencing of spike amplicons of the NTD, RBD and S1S2 junction regions are shown. Lines of short dashes connect values obtained by the chimera removed method, lines of long dashes connect values obtained by the covariant deconvolution method. All amplicons show a population shift from the reference with D614G to B.1.1.7 sequences with the appearance of P.1 sequences at the last time point. Additionally, known common polymorphisms T478K and L452R were observed from the RBD amplicons. RT-PCR for the RBD amplicon was performed in duplicate for some samples.

**Table 1 viruses-13-01647-t001:** PCR primers used to amplify spike regions for MiSeq sequencing. Upper-case lettering indicates SARS-CoV-2 sequence. Lower-case lettering indicates adapter sequence.

Region	PCR	Orienation	Primer Sequences
RBD	Primary	forward	CTGCTTTACTAATGTCTATGCAGATTC
	Primary	reverse	TCCTGATAAAGAACAGCAACCT
	Secondary	forward	acactctttccctacacgacgctcttccgatctGTGATGAAGTCAGACAAATCGC
	Secondary	reverse	gtgactggagttcagacgtgtgctcttccgatctATGTCAAGAATCTCAAGTGTCTG
NTD	Primary	forward	GTGGTGTTTATTACCCTGACAAAG
	Primary	reverse	GCTGTCCAACCTGAAGAAGA
	Secondary	forward	acactctttccctacacgacgctcttccgatctCATTCAACTCAGGACTTGTTCTT
	Secondary	reverse	gtgactggagttcagacgtgtgctcttccgatctCCAATGGTTCTAAAGCCGAAA
S1S2	Primary	forward	GCCGGTAGCACACCTTGTAA
	Primary	reverse	TGTGCAAAAACTTCTTGGGTGT
	Secondary	forward	cactctttccctacacgacgctcttccgatctCAGGCACAGGTGTTCTTACT
	Secondary	reverse	gtgactggagttcagacgtgtgctcttccgatctGTCTTGGTCATAGACACTGGTAG

## Data Availability

Raw and processed data can be accessed at https://github.com/degregory/SR_manuscript (accessed on 23 June 2021). Raw sequencing data used in this manuscript and for the MO variant monitoring project will be available pending NCBI processing under BioProject PRJNA748354.

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
