# Peer review of "Monitoring SARS-CoV-2 Populations in Wastewater by Amplicon Sequencing and Using the Novel Program SAM Refiner"

_viruses, 2021, doi:10.3390/v13081647_

Round 1

Reviewer 1 Report

This report by Gregory et al., is a novel method defined for identifying variants of SARS Cov2 from wastewater. The method is an advancement keeping in consideration the mutations in the spike protein. To me the algorithm and methodology was well defined and since as the report says that this method is in application in Missouri, this is evidently a proven approach for identifying the variants. The authors also well define the limitations which I believe with incremental development of the method would be taken into consideration at the later stages. With this I would like to congratulate the authors for sharing the method to public domain for other researchers to apply in different areas of sequencing SARS-Cov2. 

Author Response

No changes requested.

Reviewer 2 Report

With the advance of sequencing technology, high-throughput sequencing (HTS) has been successfully utilized for the SARS-CoV-2 pandemic response and genomic surveillance. HTS also has been applied to monitor the wastewater for tracking the spread of SARS-CoV-2. Gregory et al. proposed a new data analysis pipeline called SAM Refiner to remove chimeric sequences and detect SARS-CoV-2 variants including two variants of concerns. However, NGS pipelines for wastewater samples have been well established in previous studies, which should be described in Introduction section. The authors should further demonstrate SAM Refiner outperforming than those pipelines in the result. Here are suggestions to improve this manuscript.

Major comments

  1. There is no one pipeline that fits all situations. The choices of tools and details parameters depend on the sequencing technologies and platforms or specific genome structures. The authors should compare their SAM Refiner with other pipelines which have been applied on illumina data and platform, and tabulate the read coverage and depth, counts and length of contigs, and execution time using the same (benchmark) dataset.
  2. Lines 268 – 269: “Many programs exist for this task; however, we were unable to find any that provided satisfying results for our amplicon sequencing.” The author should detailly provide what programs they performed and relevant result to point out the limitations of those programs.
  3. Lines 420 – 421: “Variances that only appeared in one sequencing run and did not appear frequently in GSIAD (https://www.gisaid.org/) were considered likely PCR error and not taken into account for sequence assignment.” All of RNA viruses have high mutation rate and mutations happen randomly. Substitutions should be validated, not directly ruled out.
  4. Lines 440-442: “As that is the only sample where those covariant sequences were observed and the variances are not frequently reported in GISAID (outside of P.1 for A688V), we assigned it to the reference category.” Sequences in GISAID shows the issue of imbalanced number according to the geolocations. It is dangerous to define natural and artificial mutation by the frequency of mutations in database strains. A benchmark (or simulated) dataset are needed to estimate the performance or accuracy of detecting artificial mutations.
  5. Raw sequencing data and assembled (consensus) genome need to be deposited in SRA NCBI database and GISAID, respectively, instead of GitHub. The authors can also add descriptions in GitHub README file to tell people how to run this pipeline using the NGS data they provided and reproduce the result.
  6. The manuscript should be checked by a native English speaker.

Author Response

"There is no one pipeline that fits all situations. The choices of tools and details parameters depend on the sequencing technologies and platforms or specific genome structures. The authors should compare their SAM Refiner with other pipelines which have been applied on illumina data and platform, and tabulate the read coverage and depth, counts and length of contigs, and execution time using the same (benchmark) dataset."

While the reviewer's point is well taken about the different platforms, this is basically asking us to provide additional data to support a conclusion that we are not trying to make. We do not make comparative claims about speed or efficacy of SAM Refiner and relative to other tools, we are only claiming that SAM refiner is platform that provides the output that we find most useful for this application in a succinct manner. While head-to-head comparisons are useful when tools provide the same endpoint or function, we are unaware of any existing tool or pipeline that is equivalent to SAM Refiner. We have tried to clarify this point in lines 60-76.

"Lines 268 – 269: “Many programs exist for this task; however, we were unable to find any that provided satisfying results for our amplicon sequencing.” The author should detailly provide what programs they performed and relevant result to point out the limitations of those programs."

We have clarified in the manuscript that, to our knowledge, no existing programs/pipelines exist that provide the endpoint we desired and thus we developed SAM Refiner to provide such endpoint (lines 60-76)

"Lines 420 – 421: “Variances that only appeared in one sequencing run and did not appear frequently in GSIAD (https://www.gisaid.org/) were considered likely PCR error and not taken into account for sequence assignment.” All of RNA viruses have high mutation rate and mutations happen randomly. Substitutions should be validated, not directly ruled out."

We have clarified that polymorphisms excluded from sequence assignment to specific lineages were exclude due to the inability to validate them (lines 438-443). This analysis was not to determine whether specific polymorphisms were real or artificial.

"Lines 440-442: “As that is the only sample where those covariant sequences were observed and the variances are not frequently reported in GISAID (outside of P.1 for A688V), we assigned it to the reference category.” Sequences in GISAID shows the issue of imbalanced number according to the geolocations. It is dangerous to define natural and artificial mutation by the frequency of mutations in database strains. A benchmark (or simulated) dataset are needed to estimate the performance or accuracy of detecting artificial mutations."

We have clarified the logic we used to determine a polymorphism set could not be validated to warrant a novel lineage assignment (lines 464-465). This analysis was not to determine whether specific polymorphisms were real or artificial.

"Raw sequencing data and assembled (consensus) genome need to be deposited in SRA NCBI database and GISAID, respectively, instead of GitHub. The authors can also add descriptions in GitHub README file to tell people how to run this pipeline using the NGS data they provided and reproduce the result."

The raw sequencing data has been submitted to NCBI’s SRA database along with over 1,400 other Missouri readouts from this project (BioProject PRJNA748354). However, NCBI appears to be experiencing technical difficulties and to date only some of the raw data has been processed and is available. We have updated the manuscript to include the bioproject accession. We will also leave all the sequencing data relevant to this manuscript on github. GISAID is a repository for assembled sequences, which is not appropriate for this output. The Github Readme refers potential users to the manuscript which contains all of the information for reproducing the results.

"The manuscript should be checked by a native English speaker."

We wish the reviewer had been more specific about what portions of the manuscript that they found challenging. The authors are all native English speakers, though we admit that we found it challenging explaining some of the algorithms in an easily digestible way.

Reviewer 3 Report

  • Add more details to the abstract
  • Add a discussion section to the article. Discuss implications and other available studies/methods
  • Move lines 44-60 to discussion
  • Modify lines 40-43 and state the aim/hypothesis of the study
  • Move line 62-64 and the figure to the results
  • Was the study approved or needed ethical approval? Please provide details
  • Move part 3.4 to discussion
  • In discussion, mention the possible uses of this method in clinical settings like hospitals, dental offices, medical offices etc. In facilities that generate a big amount of waste water, ex. Dental offices, how feasible is to use such techniques as presented in this paper. Please refer to and cite these papers and similar papers in other clinical settings

Characteristics and Detection Rate of SARS-CoV-2 in Alternative Sites and Specimens Pertaining to Dental Practice: An Evidence Summary. Journal of Clinical Medicine, (2021)10(6), 1158.

Testing for COVID-19 in dental offices: mechanism of action, application and interpretation of laboratory and point-of-care screening tests. The Journal of the American Dental Association. (2021)

Author Response

"Add more details to the abstract. Add a discussion section to the article. Discuss implications and other available studies/methods. Move lines 44-60 to discussion. Modify lines 40-43 and state the aim/hypothesis of the study. Move line 62-64 and the figure to the results. Move part 3.4 to discussion"

We have added more details to the abstract and have added a short discussion section, to which the limitation section was moved. We have moved the figure but feel that moving parts of the introduction sections indicated would leave out important information for understanding the aim of the study, which we state in lines 47-49. We have reiterated relevant information from the introduction in the discussion instead.

"Was the study approved or needed ethical approval? Please provide details"

No human or animal subjects were involved in this study, so it did not need external ethical approval. Wastewater results cannot be traced back to an individual even if we wanted to.

"In discussion, mention the possible uses of this method in clinical settings like hospitals, dental offices, medical offices etc. In facilities that generate a big amount of waste water, ex. Dental offices, how feasible is to use such techniques as presented in this paper. Please refer to and cite these papers and similar papers in other clinical settings"

This manuscript is about analyzing deep sequence results from wastewater surveillance to determine viral lineages present. This is not an appropriate technique for rapid screening in a clinical setting. We can't think of a way to incorporate the discussion and references that the authors are suggesting.

Round 2

Reviewer 2 Report

Accept in present form.

Reviewer 3 Report

-